# Systemic Therapy of Gastric Cancer—State of the Art and Future Perspectives

**DOI:** 10.3390/cancers16193337

**Published:** 2024-09-29

**Authors:** Florian Lordick, Sun Young Rha, Kei Muro, Wei Peng Yong, Radka Lordick Obermannová

**Affiliations:** 1Department of Medicine (Oncology, Gastroenterology, Hepatology, Pulmonology), University of Leipzig Medical Center, Cancer Center Central Germany, 04103 Leipzig, Germany; 2Department of Internal Medicine, Yonsei Cancer Center, Yonsei University College of Medicine, Seoul 03722, Republic of Korea; 3Department of Clinical Oncology, Aichi Cancer Center Hospital, Nagoya 464-8681, Japan; 4Department of Haematology-Oncology, National University Cancer Institute, Singapore 119074, Singapore; 5Department of Comprehensive Cancer Care, Masaryk Memorial Cancer Institute, Faculty of Medicine, Masaryk University, 656 53 Brno, Czech Republic

**Keywords:** gastric cancer, esophago-gastric junction cancer, chemotherapy, immunotherapy, Her2, perioperative, neoadjuvant

## Abstract

**Simple Summary:**

A review of the latest research at PubMed and major cancer conferences was conducted to find out the current treatments for advanced stomach and esophagogastric junction cancers. In the West, neoadjuvant and perioperative chemotherapy is preferred for localized tumors. In East Asia, adjuvant chemotherapy is preferred. Studies are looking at how well immunotherapy and other drugs work in the perioperative setting. To choose the best treatment for advanced gastric cancer, including adenocarcinoma of the esophago-gastric junction, it is important to know biomarkers like HER2 expression, PD-L1 combined positive score (CPS), Claudin 18.2, and microsatellite instability (MSI). The standard first-line therapy is a combination of fluoropyrimidine and a platinum derivative. The choice of chemotherapy with antibodies depends on the biomarker. This article reviews recent clinical trial results and looks at the future of systemic therapy.

**Abstract:**

**Background:** The prognosis of patients diagnosed with locally advanced and metastatic gastric and esophago-gastric junction cancer is critical. The optimal choice of systemic therapy is essential to optimize survival outcomes. **Methods**: A comprehensive literature review via PubMed and analysis of major oncology congresses (European Society for Medical Oncology and American Society of Clinical Oncology websites) were conducted to ascertain the current status and latest developments in the systemic treatment of patients with localized or advanced gastric and esophago-gastric junction adenocarcinoma. **Results**: While neoadjuvant and perioperative chemotherapy for localized tumor stages is the preferred approach in the Western Hemisphere, adjuvant chemotherapy remains the preferred course of action in East Asia. The administration of chemotherapy, typically in the form of combinations comprising platinum and fluoropyrimidine compounds in combination with docetaxel, represents a standard of care. Investigations are underway into the potential of immunotherapy and other biologically targeted agents in the perioperative setting. To select the most appropriate therapy for advanced gastric cancer, including adenocarcinoma of the esophago-gastric junction, it is essential to determine biomarkers such as HER2 expression, PD-L1 combined positive score (CPS) (combined positive score), Claudin 18.2, and microsatellite instability (MSI). In the present clinical context, the standard first-line therapy is a combination of fluoropyrimidine and a platinum derivative. The selection of chemotherapy in combination with antibodies is contingent upon the specific biomarker under consideration. **Conclusions**: This article reviews the current state of the art based on recent clinical trial results and provides an outlook on the future of systemic therapy.

## 1. Introduction

Gastric cancer (GC) represents the fifth most prevalent malignant tumor and the fourth leading cause of cancer-associated mortality on a global scale. The incidence of gastric cancer varies geographically across the globe. The highest incidence rates are observed in Eastern Asia and Eastern Europe. In contrast, incidence rates in Northern Europe and Northern America are generally low and comparable to those observed in African regions. It is noteworthy that the incidence of gastric cancer among young adults (aged <50 years) has been on the rise in recent years in both low-risk and high-risk countries [1,2]. In addition to Helicobacter pylori infection, the occurrence of GC has been linked to genetic risk factors as well as lifestyle factors, such as alcohol consumption and smoking [3,4,5,6].

Despite the high incidence of gastric cancer (GC), most patients are unfortunately diagnosed at advanced stages with dismal prognoses due to the lack of distinguishing clinical indications. Systemic chemotherapy represents the primary treatment option for metastatic gastric cancer (mGC), with a median overall survival (OS) of approximately 12 months observed in patients treated with conventional chemotherapy [7]. Intratumoral and intertumoral heterogeneity are the common features of gastric cancer (GC), and they contribute to the poor prognosis associated with this disease. However, histological classifications are inadequate for effectively stratifying patients for personalized treatment and improving patients’ clinical outcomes. It is, therefore, evident that cutting-edge diagnostic techniques and drugs are of fundamental importance for the better characterization of molecular profiles and the identification of potential novel therapeutic targets for GC patients.

## 2. Systemic Therapy of Localized Gastric Cancer

### 2.1. Standard Perioperative Treatment

Perioperative chemotherapy is a standard of care in most parts of the Western Hemisphere for the treatment of clinical stage Ib–IVa resectable gastric cancer (GC), according to current National Comprehensive Cancer Center (NCCN) and European Society for Medical Oncology (ESMO) clinical practice guidelines [8,9,10,11]. Compared with resection alone, perioperative platinum-fluoropyrimidine-based doublet or triplet chemotherapy has been shown to increase survival by up to 15% after 5 years of follow-up [12,13]. According to the FLOT-4 trial, the FLOT regimen (fluorouracil, leucovorin, oxaliplatin, and docetaxel) is now considered the treatment of choice for patients who are eligible for a three-drug combination [14].

In contrast, in East Asian countries, especially in Japan and Korea, neoadjuvant therapy is recommended only for cancers with extensive lymph node involvement (D2 + No. 16 lymph node station), while for the majority of N+ GC and/or T2-T4 GC, postoperative chemotherapy is recommended to control residual tumor cells after curative resection [15]: The ACTS-GC trial showed that adjuvant S-1 is effective [16,17]. This trial established that postoperative S-1 monotherapy is the standard of care. The CLASSIC trial in Korea showed that a combination of capecitabine and oxaliplatin prolonged recurrence-free survival in advanced gastric cancer [18]. A combination of S-1 and docetaxel was shown to have a significant benefit in recurrence-free survival over S-1 alone in stage III gastric cancer in the JACCRO GC-07 trial [19]. S-1 plus oxaliplatin (SOX) therapy also prolonged disease-free survival compared to S-1 in the phase III ARTIST2 trial [20]. As a result, postoperative adjuvant doublet chemotherapy is now the preferred approach in East Asia when there is a higher risk of recurrence.

### 2.2. Immune Checkpoint Inhibition in Combination with Perioperative Chemotherapy

In the global phase III KEYNOTE-585, the combination of the programmed cell death-1 (PD-1) directed immune checkpoint inhibitor (ICI) pembrolizumab and perioperative chemotherapy increased the histopathological complete remission (pCR) rate by 11% (13% vs. 2%) compared to chemotherapy alone [21]. Formally, however, despite a numerical difference of 19 months, there was no statistically significant advantage in event-free survival for the addition of ICI (44.4 vs. 25.5 months, HR 0.81, *p* = 0.0198, significance threshold *p* = 0.0178). After a median follow-up of 59.9 months, the overall survival (OS) was 71.8 vs. 55.7 months. Patients were included regardless of their PD-L1 expression. In the subgroup analysis, there was a particular advantage for patients with deficient DNA mismatch repair microsatellite-instability (dMMR/MSI-High(H)) tumors. Of note, only 20% of patients were treated with the current standard perioperative chemotherapy (FLOT), while the majority of patients received a chemotherapy doublet (cisplatin and 5-FU/capecitabine), which is no longer a standard treatment in this indication. The rate of R0 resections (80% vs. 75%) and treatment-related adverse events (grades 3–4, 64% vs. 63%) were comparable in both arms. Thus, in KEYNOTE-585, the addition of pembrolizumab was safe and feasible and led to an increased pCR rate, but this did not translate into a significant prolongation of survival-related endpoints.

Two other randomized controlled trials (phase III Matterhorn and phase II Dante) confirmed the increase in pCR rates with the addition of ICIs to neoadjuvant therapy. However, survival-related endpoints have not yet been published from these two trials [22,23].

ATTRACTION-5 was a randomized, placebo-controlled trial conducted in East Asia. Patients with stage IIIA-C gastric or esophago-gastric junction (EGJ) adenocarcinoma after gastrectomy with D2 or more extensive lymph-node dissection were randomly assigned to receive either nivolumab plus chemotherapy or placebo plus chemotherapy. The investigational treatment, either nivolumab or placebo, was given intravenously for 30 min once every three weeks. The relapse-free survival hazard ratio was 0.90 (*p* = 0.44). This trial did not show that adding nivolumab to post-surgery treatment helps Asian patients with advanced gastric or EGJ cancer. [24].

Deficient DNA mismatch repair (dMMR)/MSI-H) cancers are a biologically distinct subset characterized by an overall better prognosis after resection and presumably no benefit from perioperative or adjuvant chemotherapy [25]. The high responsiveness of advanced dMMR/MSI-H GC to ICI [26] makes this subset an interesting target for neoadjuvant immunotherapy. All the above-mentioned studies (KEYNOTE-585, Matterhorn, and Dante) reported significantly higher response rates for dMMR/MSI-H GC in the ICI arms [22,23].

The French GERCOR Phase II NEONIPIGA study evaluated a perioperative treatment concept with ICI alone for 32 dMMR/MSI-H tumors. Following neoadjuvant therapy with ipilimumab and nivolumab for 12 weeks, a pCR rate of 59% was achieved in 29 patients, all of whom underwent R0 resection. Nivolumab was also administered adjuvantly in 23 cases. Three patients were not resected (patient request in 2 cases, metastases at inclusion in one case) and showed clinically complete remission with tumor-free biopsies and CT imaging up to a follow-up period of 14.9 months [27]. Similar results were obtained for 15 patients with dMMR/MSI-H gastric carcinomas and AEG tumors in the phase II INIFINITY study [28]. After neoadjuvant therapy with tremelimumab and durvalumab for 12 weeks, pCR was achieved in 60% of patients. Despite the small number of cases, both studies impressively demonstrate that ICI is a promising treatment concept in the molecular subset of dMMR/MSI-H GCs, which may replace perioperative chemotherapy for resectable GCs in the future. In view of the high pCR rates, the question also arose whether surgical resection of the primary tumor could be omitted in this subgroup. An organ-preserving approach with ICI is being evaluated in Cohort 2 of the INFINITY trial [28].

In conclusion, neither neoadjuvant nor adjuvant immunotherapy has yet been shown to improve survival in patients with GC. However, randomized phase III trials are ongoing, and the results of survival-related endpoints are still pending. The dMMR/MSI-H subgroup is most likely to benefit from the addition or even replacement of perioperative chemotherapy with ICIs [21,22,23,24,25,26,27,28].

### 2.3. Perioperative HER2-Targeted Therapy in Combination with Perioperative Chemotherapy

The single-arm phase II HERFLOT study showed for the first time that the addition of trastuzumab to perioperative chemotherapy with FLOT in GC achieved a pCR rate of 21.4% [29]. A total of 56 patients were included, 40 of whom had EGJ tumors. In the HERFLOT study, only intestinal subtypes benefited from trastuzumab compared to diffuse subtypes. Tumor location did not affect results.

The randomized-controlled German AIO-PETRARCA trial and multicenter European EORTC-INNOVATION study demonstrated higher pCR rates when HER2-targeted therapy was added to perioperative chemotherapy [30,31]. As both studies recruited a limited number of patients and follow-up periods are still short, the effects on survival-related outcomes are still uncertain. However, it was demonstrated that the combination of FLOT with two HER2 inhibitors (Trastuzumab + Pertuzumab) is too toxic, while the combination of trastuzumab and FLOT was feasible and promising [31]. Better Her-2 targeting strategies with reasonable tolerability need to be evaluated and should be further investigated in a properly powered randomized-controlled phase III study.

## 3. Systemic Therapy of Advanced Gastric Cancer

### 3.1. Current Status

The range of systemic treatment options for metastatic or unresectable gastric cancer has been expanded in recent years, representing a significant advancement in the field of oncology. Until recently, the standard of care for human epidermal growth factor receptor 2 (HER2)-negative tumors was a first-line therapy comprising a fluoropyrimidine, typically 5-fluorouracil (5-FU), capecitabine, or S-1, in combination with a platinum derivative. At that time, the only targeted therapy available for first-line treatment of GC was the combination of trastuzumab and chemotherapy for HER2-positive tumors, as evidenced by data from the randomized controlled phase III ToGA study [32].

Recently, a number of novel compounds have been approved for use based on evidence of efficacy in molecularly defined subgroups. The use of these drugs is contingent upon the detection of defined biomarkers. The new approvals include ICIs which target the PD-1/PD-L1 checkpoint, the antibody-drug conjugate (ADC) trastuzumab deruxtecan (T-DXd), which targets HER2, and Zolbetuximab which binds to Claudin18.2. This exciting development will be outlined and referenced in the following paragraphs that will focus on the newly approved drugs, with particular emphasis on the data that informed their approval by the European Medicines Agency (EMA) and the US Food and Drug Administration (FDA). Additionally, an overview of the compounds that have demonstrated activity and efficacy in recent studies will be provided.

### 3.2. Biomarkers

Prior to initiating first-line systemic therapy for advanced gastric cancer, it is mandatory to determine the status of biomarkers (Figure 1, Figure 2), as the selection of the optimal treatment combination and sequence is contingent upon the result. The four established markers to guide first-line therapy are HER2, PD-L1, Claudin18.2, and MMR/MSI status.

A tumor is classified as HER2-positive if either the immunohistochemistry (IHC) score is 3+ or the IHC score is 2+, and the in situ hybridization (ISH) is positive with a *HER2/CEP17* ratio of ≥2. A notable characteristic of GC is the frequent intratumoral heterogeneity of HER2 expression. There may be a variation in HER2 status between different tumor sites or even within the same tumor site, which can hamper the efficacy of HER2-directed therapy [33].

The diagnosis of dMMR or MSI-H is achieved through immunohistochemical staining of the DNA mismatch repair proteins or, alternatively, through genetic analysis via polymerase chain reaction (PCR) or next-generation sequencing (NGS) [34]. The IHC test demonstrates a high degree of correlation with the MSI PCR result and can be conducted rapidly and inexpensively [35].

The immunohistochemical (IHC) determination of the marker PD-L1 is also a standard procedure. In adenocarcinoma of the stomach and EGJ, the PD-L1 status according to the combined positive score (CPS) is currently the decisive biomarker; the tumor proportion score (TPS) plays no role. The CPS takes into account the staining of tumor cells and immune cells (lymphocytes and macrophages) in the tumor stroma, whereas the TPS only considers the staining of tumor cells. With accumulating data, higher PD-L1 expression is related to more immunogenic tumor microenvironment and more responsiveness to ICI. Based on different studies, there remains the debate regarding the proper cut-off for clinical practice [36].

It seems reasonable to anticipate that the determination of Claudin18.2 will become increasingly important in the future. Claudin18.2 is a tight junction protein against which the antibody zolbetuximab is directed. The phase II FAST study demonstrated that an effect of Zolbetuximab could only be anticipated in cases where strong expression (IHC score 2+ or 3+) was observed in more than 70% of tumor cells [37]. Consequently, the recently published phase III studies exclusively included patients with a claudin18.2 tumor expression of ≥75% of tumor cells [38,39].

With the increasing availability of zolbetuximab, a question will arise as to which targeted therapy should be selected in cases where multiple molecular markers are positive. A review of 408 samples revealed an overlap between claudin18.2 and PD-L1 positivity (defined as CPS ≥ 5) in approximately 7% of the examined tumors [40]. Co-expression of Claudin 18.2 and PD-L1 was reported in 13–22% of patients from GLOW and SPOTLIGHT [38,39].

### 3.3. Principles of First-Line Chemotherapy

For a detailed overview of the principles of chemotherapy, please refer to the most recent ESMO guidelines [10]. The fundamental principle of chemotherapy is a double combination of a platinum derivative (either oxaliplatin or cisplatin) and a fluoropyrimidine (either 5-FU, capecitabine, or S-1, provided that oral tablet absorption is not a problem). The triple combination, supplemented by docetaxel, should only be used in the metastatic situation in justified cases. These cases present with a strong remission pressure, vital-threatening metastasis, and a good to very good performance status. Oxaliplatin has established itself as the standard due to its generally better tolerability. In contrast, due to the higher rate of side effects such as nausea and vomiting, nephrotoxicity, and ototoxicity, cisplatin is usually only used in cases of severe intolerance to oxaliplatin. In elderly or fragile patients, dose-reduced chemotherapy should be used, which, according to the GO2 study, contributed to comparable disease control as full-dose chemotherapy [41].

### 3.4. Microsatellite Stable (MSS) Tumors (Or Proficient Mismatch Repair, pMMR), HER2-Negative

This group represents the largest molecular subgroup among GCs. Two immune ICIs, nivolumab and pembrolizumab, have been approved for first-line treatment by EMA and FDA in combination with chemotherapy. Other regions have access to additional or alternative ICIs. Following the recommendations set forth by both the ESMO and NCCN guidelines, the use of ICIs is contingent upon the PD-L1 status of the tumor [9,10] (Figure 2).

The combination of fluoropyrimidine/platinum derivative and nivolumab was approved based on the findings of the CHECKMATE-649 study, which indicated that patients with a PD-L1 CPS ≥ 5 exhibited better overall survival [42]. In light of the findings from the KEYNOTE-859 study [43], the combination of fluoropyrimidine/platinum derivative and pembrolizumab was also granted approval for first-line use in patients with a PD-L1 CPS ≥ 1. FDA and some other agencies around the world have decided to grant access regardless of the PD-L1 status of the tumor, although the efficacy of ICIs in the case of low or no PD-L1 expression appears to be small or even negligible [44]. The risk-benefit and the cost-benefit ratio are probably unfavorable in case of a negative PD-L1 status. During ICI therapy, close attention should be paid to immune-mediated adverse events, and ESMO guidelines should be available for monitoring and management [45].

### 3.5. HER2-Positive Tumors

The determination of PD-L1 status has now been established as a criterion for the selection of therapy in the initial treatment of HER2-positive tumors. (Figure 2). In the case of a PD-L1-negative tumor with a CPS of 0, the combination of trastuzumab with a chemotherapy regimen comprising a fluoropyrimidine and a platinum derivative remains the preferred treatment option, as demonstrated in the ToGA study [32]. The use of substances directed against HER2 carries an inherent risk of left heart failure, which necessitates the implementation of regular ECG and echocardiography monitoring.

A significant advancement has been made in the understanding of HER2 positivity and PD-L1 positivity (CPS ≥ 1). The data from the KEYNOTE-811 study resulted in the approval of the combination of fluoropyrimidine/platinum derivative with trastuzumab and pembrolizumab in Europe in 2023 [46]. FDA had granted approval earlier, in 2021, based solely on the data indicating an improved response rate. The progression-free survival data have now been published, and on the basis of these findings, the EMA has restricted approval to the CPS ≥ 1 group, as the benefit from the ICI is restricted to this patient cohort.

The antibody-drug conjugate T-DXd received approval for the second or third-line treatment of HER2-positive GCs. The approval is based on the findings of two phase II clinical trials. The Destiny-Gastric-01 trial was conducted in East Asia and compared T-DXd with standard chemotherapy as a third-line or beyond setting in a randomized manner and demonstrated significantly improved overall response rate and overall survival in the T-DXd group [47]. In the Destiny-Gastric-02 trial, patients of second-line treatment in Europe and the USA were treated with T-DXd. Treatment outcomes were consistent with those reported in Destiny-Gastric-01 [48]. It is noteworthy that all patients had previously undergone treatment with a trastuzumab-containing regimen. Moreover, Destiny-Gastric-02 necessitated a rebiopsy of the tumor lesion to substantiate sustained HER2 positivity, as this is lost in up to 60% of cases under HER2-targeted therapy [49].

Patients treated with T-DXd should be meticulously observed for indications of interstitial lung disease (ILD). Even in the event of the detection of asymptomatic abnormalities on chest CT scans, the administration of T-DXd should be terminated. In the event of the emergence of symptoms, the administration of steroids is recommended, and the drug in question should be permanently discontinued [50]. The therapeutic implications of a low-positive HER2 finding (Her2-low) in gastric carcinoma remain uncertain and are the subject of ongoing investigation [51].

### 3.6. dMMR/MSI-H Tumors

Data from multiple Phase III trials have evaluated the subset of MSI-H tumors and demonstrated a clear benefit of ICI treatment compared to chemotherapy alone [28]. The combination of a checkpoint inhibitor with chemotherapy is recommended for all MSI-H tumors, as MSI-H GCs often do not respond to chemotherapy alone, and in the event of progression and clinical deterioration, there may be no further treatment options. The question of whether an ICI alone is equivalent to the combination of an ICI and chemotherapy remains open. In the Japanese No Limit study, the chemo-free strategy with a combination of nivolumab and low-dose ipilimumab demonstrated high and robust efficacy with good tolerability in MSI-H GC patients [52]. A favorable efficacy outcome of ipilimumab and nivolumab in the MSI-H subset of patients was reported in CHECKMATE 649 [53]. In KEYNOTE 062, the combination of pembrolizumab and chemotherapy had better progression-free survival than pembrolizumab alone, but the number of cases is small [54]. If an MSI-H tumor has progressed under chemotherapy alone without an ICI being used in the first line, pembrolizumab has shown activity in the second line of therapy and should be prescribed [55].

### 3.7. Claudin18.2 Positive Tumors

Two positive Phase III studies of zolbetuximab in combination with chemotherapy for first-line claudin18.2-positive gastric or EGJ cancer were published in 2023. In the SPOTLIGHT study, zolbetuximab was combined with modified FOLFOX6, and in the GLOW study, zolbetuximab was combined with capecitabine/oxaliplatin. Both studies included only patients with HER2-negative tumors and required moderate or strong Claudin18.2 expression (IHC score 2+ or 3+) on ≥75% of tumor cells [38,39]. Both studies showed improved progression-free and overall survival with the combination of zolbetuximab and chemotherapy compared to chemotherapy alone. Nausea and vomiting were significantly more common with zolbetuximab than in the comparator arm. In many patients, nausea and vomiting occur mainly during the first zolbetuximab infusions and then decrease in intensity [56]. In the Phase II ILUSTRO trial, zolbetuximab alone showed significantly lower activity than in combination with chemotherapy [57], so this approach will not be pursued further. Claudin18.2 is also a highly investigated target for other therapeutic approaches, such as combinations with immune checkpoint inhibitors, antibody-drug conjugates, bispecific antibodies, cellular agents, and vaccines [58].

### 3.8. FGFR2b Positive Tumors

The efficacy of bemarituzumab, an antibody against fibroblast growth factor receptor 2 (FGFR2b), in combination with chemotherapy, is currently being investigated in randomized controlled phase III trials [59]. In the randomized phase II FIGHT study, overall response rate and survival-related outcomes were prolonged in the group of patients with FGFR2b-positive tumors (IHC 2+/3+) who received bemarituzumab and first-line chemotherapy [60].

### 3.9. Second- and Third-Line Therapy of Gastric Cancer

There have been few innovations in clinical practice in the post-progression lines of therapy other than the aforementioned T-DXd for HER2-positive carcinomas and pembrolizumab for MSI-H carcinomas. The established second-line therapy is ramucirumab and paclitaxel, based on the results of the RAINBOW trial [61]. Initial results from the phase II RAMIRIS trial showed an advantage for FOLFIRI/ramucirumab in the taxane-pretreated arm, and this combination is currently being further evaluated in a phase III trial [62].

In the third line, irinotecan, paclitaxel, docetaxel, or trifluridine/tipiracil are recommended, depending on the patient’s prior therapy [10]. The combination of regorafenib plus nivolumab has demonstrated promising data, and a randomized trial of this combination is currently underway [63].

In the event that the patient’s general condition is favorable and the disease is unresponsive to standard treatment, the tumor should be examined using next-generation sequencing (NGS), and the findings should be discussed on a molecular tumor board. Following this discussion, the question of whether targeted therapies in off-label use or molecularly stratified studies are an option will be addressed. Several rare alterations might be expected (Table 1), but more information is needed. Although fusions involving NTRK or RET are uncommon in gastric carcinoma, they may be amenable to treatment with an approved therapy targeting NTRK (larotrectinib, entrectinib) or RET (selpercatinib). Given their low frequency, these cases should not be overlooked.

## 4. Conclusions

For localized and resectable gastric cancer, perioperative chemotherapy, preferably using the FLOT regimen, is the standard of care in the Western Hemisphere. This is recommended for all localized cancers beyond early stages. This standard is not universally adopted in East Asia, where adjuvant chemotherapy is still the preferred approach. S1 or platinum-fluoropyrimidine doublets or a fluoropyrimidine-taxane combination is considered the standard of care. In the case of bulky nodal disease or borderline resectable locally advanced gastric cancer, a neoadjuvant approach is widely recommended in East Asia.

In order to select the most appropriate therapy for patients with advanced gastric cancer, including adenocarcinoma of the EGJ, it is essential to determine the four key criteria: HER2 expression, PD-L1 expression by CPS, Claudin 18.2 status, and microsatellite instability. In the present clinical context, the standard first-line therapy is a combination of fluoropyrimidine and a platinum derivative. The choice of chemotherapy combined with antibodies is dependent on the specific biomarker under consideration. In the case of HER2-positive tumors, a combination of chemotherapy and trastuzumab is recommended for those with PD-L1-negative tumors (CPS 0). In cases where the PD-L1 CPS is equal to or greater than 1, the recommended course of action is to supplement the existing therapy with pembrolizumab. In the case of Claudin-18.2-positive tumors, Zolbetuximab should be added to chemotherapy, contingent upon its availability and potential overlap with a positive PD-L1 CPS. In such instances, physicians and patients must determine whether to add an ICI or Zolbetuximab. For microsatellite-stable PD-L1 positive tumors (MSS/pMMR), immune checkpoint inhibitors (ICIs) are available and should be prescribed depending on the PD-L1 score. A combination of chemotherapy and an ICI is recommended for the treatment of dMMR/MSI-H tumors. New biomarkers such as FGFR2b expression and new technologies such as bispecific antibodies, antibody-drug conjugates (ADCs), or cellular therapies may enter the field and enhance the therapeutic armamentarium for patients with advanced gastric cancer (GC).

## Figures and Tables

**Figure 1 cancers-16-03337-f001:**
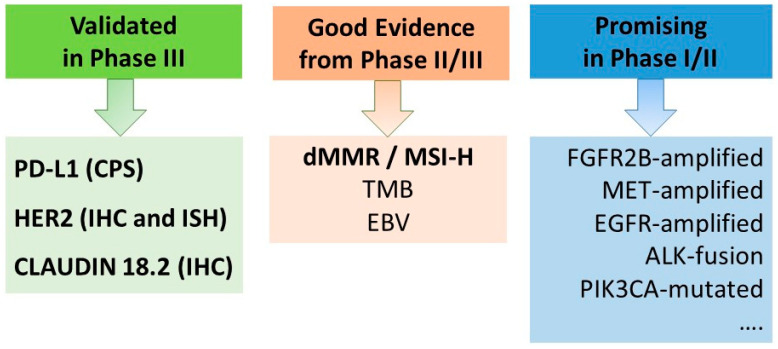
Biomarkers to select treatment for patients with advanced gastric cancer. ALK, anaplastic lymphoma kinase; EBV, Epstein-Barr-Virus; EGFR, epidermal growth factor receptor; dMMR, deficient DNA mismatch repair; FGFR2b, fibroblast growth factor receptor 2b; fus, fusion; HER2, human epidermal growth factor receptors-2; MET, mesenchymal-epithelial transition factor; MSI-H, microsatellite instability-high; PIK3CA, phosphatidylinositol-4,5-bisphosphate 3-kinase catalytic subunit alpha; PD-L1, programmed cell death 1; TMB, tumor mutational burden.

**Figure 2 cancers-16-03337-f002:**
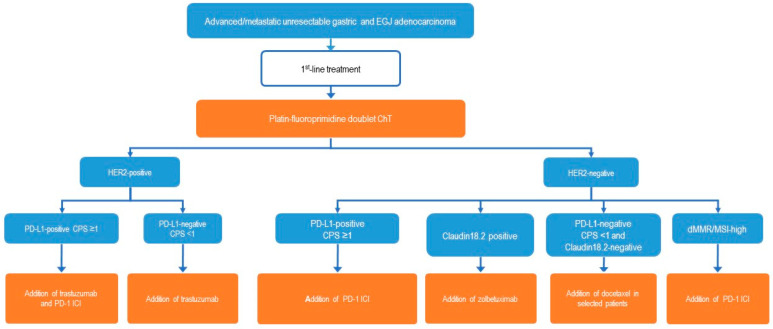
First-line treatment algorithm for patients with advanced gastric cancer. dMMR, deficient DNA mismatch repair; HER2, human epidermal growth factor receptos-2; MSI-H, microsatellite instability high; PD-L1, programmed cell death 1. Modified according to European Society for Medical Oncology Gastric Cancer Guidelines [10].

**Table 1 cancers-16-03337-t001:** Molecular alterations and potential drugs.

Molecular Alteration	Frequency	Drug	Reference
*ALK* fusion/expression	1%/8%	Alectinib, Lorlatinib	[64,65]
*EGFR* amp	6–10%	Afatinib, Cetuximab	[65,66,67,68,69,70,71]
*FGFR2* amp	2–11%	Erdafitinib, Pemigatinib, Regorafenib, Futibatinib	[60,67,72,73,74,75,76,77]
*FGFR1* amp	2%	Erdafitinib, Pemigatinib, Regorafenib	[60,66,67,72,73,74,75]
*FGFR3* amp	2%	Erdafitinib, Pemigatinib, Regorafenib	[60,66,72,73,74,75,76,77,78]
HRD	7–12%	Olaparib	[79,80,81,82]
*KRAS G12C* mut	1%	Sotorasib, Adagrasib	[83,84]
*MET* amp	2–11%	Crizotinib, Cabozantinib, Savolitinib	[67,82,85,86,87]
*PIK3CA* mut/amp	3.5%	Alpelisib	[86]
**Immuno markers**			
EBV+ in PD-L1-	<1–2%	ICI	[67,87,88,89,90,91]
TMB-high in MSS GC	20%	ICI	[92,93]

**Legend:** ALK, anaplastic lymphoma kinase; amp, amplification; EBV+, Epstein-Barr-Virus-positive; EGFR, Epidermal Growth Factor Receptor; FGFR, fibroblast growth factor receptor; HRD, homologous recombination deficiency; ICI, immune checkpoint inhibitor; IHC; immunohistochemistry; KRAS, Kirsten rat sarcoma virus; MET, Methionine; MSS, microsatellite stable; mut, mutation; PD-L1-, programmed Cell Death Ligand-1 negative; PIK3CA, phosphatidylinositol-4,5-bisphosphate 3-kinase, catalytic subunit alpha; TMB, tumor mutational burden.

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
