# Peer review of "Systemic Therapy of Gastric Cancer—State of the Art and Future Perspectives"

_cancers, 2024, doi:10.3390/cancers16193337_

Round 1

Reviewer 1 Report

Comments and Suggestions for Authors

In this article “Systemic Therapy of Gastric Cancer – State of the Art and Future Perspectives.” Authors have done a comprehensive literature review and analysis of major oncology congresses to ascertain the current status and latest developments in the systemic treatment of patients with localized or advanced gastric and esophago-gastric junction adenocarcinoma. They also tried to mention the preferred course of action in East Asia and in the Western Hemisphere. The administration of chemotherapy, typically in the form of combinations comprising platinum and fluoropyrimidine compounds in combination with docetaxel, represents a standard of care. Investigations are underway into the potential of immunotherapy and other biologically targeted agents in the perioperative setting. To select the most appropriate therapy for advanced gastric cancer, including adenocarcinoma of the esophago-gastric junction, it is essential to determine biomarkers such as HER2 expression, PD-L1 combined positive score (CPS) (combined positive score), Claudin 18.2, and microsatellite instability (MSI).

Authors have given extensive data about the subject. However, authors should would include a flow chart explaining protocols/congress/trial/staging/patient survival to simplify the experimental regimen/strategies/state of art and Future Perspectives for readers to understand and efficiently use the information in their research questions.

Please find my comments about the manuscript.

1.     Please remove “Palliative” in key words as authors have not mentioned anything about palliative care. But authors should include palliative care as that is also included as the standard of care and what treatment regimen is followed for those patients.

2.     Authors should make a table for all the individual sections to explain staging/regimen/therapies/survival rate/congresses/ as per different congresses with dates. The year should be mentioned about trials/congresses to understand it better chronologically.

3.     In section “3.1. Current status” there is no reference for last but one paragraph in this section. Please add appropriate references here and throughout.

4.     In section “2.2. Immune checkpoint inhibition in combination with perioperative chemotherapy” there is no reference given at the end of paragraph in which authors have given immensely important data.

5.      References should not be more than 10 years old. Please update the reference section. Like some references are old as 2006.

6.       Please include a flow chart explaining protocols to simplify experimental regimen/state of art for readers to understand.

7.     Abbreviations need to be elaborated when using for the first time like “ in the section - Immune checkpoint inhibition in combination with perioperative chemotherapy section”-HR is used. Similarly, there are many more places where abbreviations have been used and not elaborated. Please expand.

Please include all the changes in the revised manuscript.

Author Response

Reviewer 1

  1. Please remove “Palliative” in key words as authors have not mentioned anything about palliative care. But authors should include palliative care as that is also included as the standard of care and what treatment regimen is followed for those patients.

Answer: done

  1. Authors should make a table for all the individual sections to explain staging/regimen/therapies/survival rate/congresses/ as per different congresses with dates. The year should be mentioned about trials/congresses to understand it better chronologically.

Answer: due to word count restrictions we could not follow this suggestion

  1. In section “3.1. Current status” there is no reference for last but one paragraph in this section. Please add appropriate references here and throughout.

 Answer: Section 3.1. amended

  1. In section “2.2. Immune checkpoint inhibition in combination with perioperative chemotherapy” there is no reference given at the end of paragraph in which authors have given immensely important data.

 Answer: references added

  1. References should not be more than 10 years old. Please update the reference section. Like some references are old as 2006.

 Answer: from a scientific viewpoint, this comment is not appropriate. Why shouldn’t we refer to important important literature which is older than 10 years? We strongly disagree with this kind of looking at our article.

  1. Please include a flow chart explaining protocols to simplify experimental regimen/state of art for readers to understand.

Answer: we think that the article is easy to understand and does not need a flowchart. What should this flowchart show?

  1. Abbreviations need to be elaborated when using for the first time like “ in the section - Immune checkpoint inhibition in combination with perioperative chemotherapy section”-HR is used. Similarly, there are many more places where abbreviations have been used and not elaborated. Please expand.

       This comment is not appropriate. We have worked on this article with a lot of care.

Reviewer 2 Report

Comments and Suggestions for Authors

Excellent review, which should help, doctors starting in the field to have a clear and uptodate overview of the field. As the authors mention they have used unpublished data from congresses, I believe adding VESTIGE and RENAISSANCE concept/data will deepen the reader's understanding.

Author Response

Reviewer 2

Excellent review, which should help, doctors starting in the field to have a clear and uptodate overview of the field. As the authors mention they have used unpublished data from congresses, I believe adding VESTIGE and RENAISSANCE concept/data will deepen the reader's understanding.

Answer: thank you for this positive comment. VESTIGE has not been fully published yet – we decided not to add this abstract with preliminary data to this review article as data may not be mature yet. Renaissance is about surgical management of oligometastatic disease. This topic clearly goes beyond the scope of this invited article. Please discuss withe the guest editors Takeshi Sano and Stefan Mönig.

Reviewer 3 Report

Comments and Suggestions for Authors

The review "Systemic Therapy of Gastric Cancer – State of the Art and Future Perspectives" presents a summary of important information from the literature regarding the systemic treatment of gastric cancer.

Recommendations:

1.     The introduction is missing!

2.     Information regarding how the studies were selected is also absent.

3.     Figures 1 and 2 are of poor quality and appear unprofessional. They need to be redone.

4.     At least 2 tables should be created to include the main studies presented in this review and their results. There is a lot of mixed information, from old studies to clinical trials, etc. The information needs to be organized more clearly.

5.     Discuss the importance of prevention in gastric cancer with a focus on two elements:

-       MicroRNA – perhaps the most modern diagnostic method, which is gaining more ground – I recommend this article: 10.3390/ijms25147898

-       Imaging – endoscopic ultrasound, non-radiating, which can also make a differential diagnosis with very rare pathologies – I recommend this article: 10.3390/diagnostics14070675

6.     Add a discussion section in which you present the main discrepancies in the literature.

7.     Conclusions are missing! The authors' contribution is also missing!

8.       Include information about the limitations of the study.

Author Response

Reviewer 3

  1. The introduction is missing!

Answer: added

  1. Information regarding how the studies were selected is also absent.

Answer: added

  1. Figures 1 and 2 are of poor quality and appear unprofessional. They need to be redone.

    They are technically good. Not our fault that the Editorial Manager reproduces them in poor quality
  2. At least 2 tables should be created to include the main studies presented in this review and their results. There is a lot of mixed information, from old studies to clinical trials, etc. The information needs to be organized more clearly.

    The information is excellently organized. A further table is not needed
  3. Discuss the importance of prevention in gastric cancer with a focus on two elements:

-       MicroRNA – perhaps the most modern diagnostic method, which is gaining more ground – I recommend this article: 10.3390/ijms25147898

-       Imaging – endoscopic ultrasound, non-radiating, which can also make a differential diagnosis with very rare pathologies – I recommend this article: 10.3390/diagnostics14070675

The topic of the article is systemic therapy.

  1. Add a discussion section in which you present the main discrepancies in the literature.

    This comment is very generic. Everything is discussed very well
  2. Conclusions are missing! The authors' contribution is also missing!

    Answer: added
  3. Include information about the limitations of the study.

    This is a review article and not an original study

Reviewer 4 Report

Comments and Suggestions for Authors

1.This paper is a well-written review article summarizing the latest treatment trends in gastric cancer.

2.Figure 2 lacks readability. It may need adjustments such as increasing the font size to improve understanding.

3.There is no conclusion section.

4.The author contribution section has not been written.

5.Recently, interest in complementary and alternative medicine has been increasing. It seems necessary to include a discussion on the efficacy and methods of gastric cancer treatment through complementary and alternative medicine.

6.Is it possible to regulate gastric cancer through ion channel modulation? If so, please supplement the paper with relevant content.

Author Response

Reviewer 4

1.This paper is a well-written review article summarizing the latest treatment trends in gastric cancer.

Answer thank you

2.Figure 2 lacks readability. It may need adjustments such as increasing the font size to improve understanding.

Answer: The figures are technically good. Not our fault that the Editorial Manager reproduces them in poor quality

3.There is no conclusion section.

Answer: added

4.The author contribution section has not been written.

Answer: added

5.Recently, interest in complementary and alternative medicine has been increasing. It seems necessary to include a discussion on the efficacy and methods of gastric cancer treatment through complementary and alternative medicine.

Answer: out of the scope of this article

6.Is it possible to regulate gastric cancer through ion channel modulation? If so, please supplement the paper with relevant content.

Answer: out of the scope of this article